# Single-Cell RNA Sequencing in Parkinson’s Disease

**DOI:** 10.3390/biomedicines9040368

**Published:** 2021-04-01

**Authors:** Shi-Xun Ma, Su Bin Lim

**Affiliations:** 1Institute for Cell Engineering, Johns Hopkins University School of Medicine, Baltimore, MD 21205, USA; shixun625@jhmi.edu; 2Department of Biochemistry and Molecular Biology, Ajou University School of Medicine, Suwon 16499, Korea

**Keywords:** Parkinson’s disease, single-cell RNA sequencing, bioinformatics

## Abstract

Single-cell and single-nucleus RNA sequencing (sc/snRNA-seq) technologies have enhanced the understanding of the molecular pathogenesis of neurodegenerative disorders, including Parkinson’s disease (PD). Nonetheless, their application in PD has been limited due mainly to the technical challenges resulting from the scarcity of postmortem brain tissue and low quality associated with RNA degradation. Despite such challenges, recent advances in animals and human in vitro models that recapitulate features of PD along with sequencing assays have fueled studies aiming to obtain an unbiased and global view of cellular composition and phenotype of PD at the single-cell resolution. Here, we reviewed recent sc/snRNA-seq efforts that have successfully characterized diverse cell-type populations and identified cell type-specific disease associations in PD. We also examined how these studies have employed computational and analytical tools to analyze and interpret the rich information derived from sc/snRNA-seq. Finally, we highlighted important limitations and emerging technologies for addressing key technical challenges currently limiting the integration of new findings into clinical practice.

## 1. Introduction

Parkinson’s disease (PD) is the second most common neurodegenerative disorder and affects over 1% of the population over the age of 60 [1]. In the United States, approximately 1.04 million individuals were diagnosed with PD in 2017, and the number of persons living with PD is expected to double by 2040, affecting people of all ages, races, and ethnicities [2,3]. In 2017, the estimated total economic burden of PD in the US was $51.9 billion, including direct medical costs and indirect and non-medical costs, such as loss in wages and social productivity; the total economic burden is expected to increase to about $79.1 billion in 2037 [3]. As the global incidence and prevalence of PD is increasing, there have been worldwide efforts to combat PD and understand the disease at the molecular level by leveraging advanced technologies.

PD is characterized by a loss of dopaminergic neurons (DaNs) in the substantia nigra pars compacta (SNpc), resulting in motor symptoms such as rigidity, postural instability, tremor at rest, and bradykinesia [4]. While dopaminergic drugs and deep-brain stimulation alleviate the symptoms and form the mainstay of PD treatment at present, they do not get to the root causes of the disease and fail to accomplish disease modification in PD patients. The other major pathologic feature of PD is the accumulation of small and complex structures called Lewy bodies (LBs), which are enriched in aggregated forms of α-synuclein (α-syn), including fibrils [5]. Yet, the processes that govern α-syn fibrillization and the biogenesis of the LBs remain poorly unknown. Further, only 15% of PD patients report a family history of PD symptoms, with varying genetic predispositions, while the remaining 85% of the PD populations are classified as sporadic PD, which do not harbor an interpretable genetic cause [6].

Single-cell and single-nucleus RNA sequencing (sc/snRNA-seq) technologies have become instrumental for assessing heterogeneous cell types and for reconstructing temporal and spatial dynamics of complex tissues [7,8,9]. With the advent of DNA barcode and combinatorial indexing strategy, up to millions of cells or nuclei can now be sequenced from a single experiment, enabling ultra-high throughput sc/snRNA-seq of samples across different tissues and in the context of a wide range of diseases. While past efforts in deconvoluting the complex nature of neural circuits have been largely ineffective with bulk assays of the average composition of a brain tissue, single-cell sequencing technologies have the advantage of characterizing the cellular heterogeneity that governs the key aspect of neurobiology. Without the need for selective cell purification, sc/snRNA-seq technologies, including transcriptomic (i.e., abundance of RNA molecules) and epigenomic (i.e., chemical and physical modifications of DNA and histone proteins) assays, measure RNA or DNA from individual cells [10]. While sc/snRNA-seq technologies, including multi-omic techniques covering multiple different modalities, have been extensively leveraged in neuroscience, they are just beginning to be applied in studies of PD, particularly with human postmortem brain tissues. Using animal and human in vitro models of PD and parkinsonism, these assays have transformed our understanding of the cellular composition and diversity of neuronal and glial cell type identities in the developing mouse and human brain and have identified their functional role in the DaN degenerative process underlying PD. Sc/snRNA-seq data derived from studies examined below can represent a possible starting point for the development of tools for targeted functional studies, connecting PD-specific transcriptomic signatures with spatial context and physiology.

## 2. Animal and Human In Vitro Models of PD and Parkinsonism

While most PD is idiopathic, genetic models of PD have provided deep insights into the more common sporadic form of the disease. Recent meta-analyses of genome-wide association studies (GWAS) in PD have identified novel loci for disease risk and the genetic variants that deterministically drive the disease or alter risk. These studies have provided useful biological insights into the pathophysiology of PD and the opportunity to develop animal models, which are, in many cases, excellent surrogates for in vivo whole brain systems [11,12]. Mutations in SNCA (α-syn) and LRRK2 (leucine-rich repeat kinase 2) cause autosomal dominant PD, while mutations in PINK1 (PTEN-induced putative kinase 1), PRKN (parkin), ATP13A2 (ATPase 13A2), DJ-1 (protein DJ-1), FBXO7 (F- box protein 7), and PLA2G6 (A2 phospholipase group VI) genes cause autosomal recessive PD [13]. Novel susceptibility genes associated with an increased risk of developing PD have also been identified, including NR4A2 (Nurr1, nuclear receptor superfamily protein), SNCAIP (synphilin-1), APOE (apolipoprotein E), MAPT (tau protein), GBA (b-glucocerebrosidase), and COMT (catechol-O-methyl transferase) [13,14,15].

Genetic-based models of PD and parkinsonism have involved familial PD-associated mutant forms of SNCA (A53T, A30P, E46K), overexpression or knock-in mutations of LRRK2 (G2019S, R1441C/G), or deletion or knock-out of PRKN, PINK1, and/or DJ-1. Although these genetic forms of PD models exhibited substantial neurodegeneration and phenocopy human PD to some extent, they often lacked measurable loss of DaNs and the resulting motoric dysfunctions [16,17,18], posing challenges for using current models to inform therapeutic intervention. On the other hand, conditional, temporal, and/or cell type-specific overexpression of mutant SNCA [11,19] and overexpression of adenoviral (AAV)-mediated transduction of mutant LRRK2 (G2019S) [20,21] led to neurodegeneration of DaNs. Similarly, loss of function models of PRKN, PINK1, or DJ-1 result in loss of DaNs in mice and rats [11,22,23,24], serving as robust experimental systems to understand molecular and cellular mechanisms leading to brain cell dysfunction and degeneration and the effect of PD-causing mutations on these processes, in PD.

PD is a prion-like disorder characterized by the spread of pathologic α-syn from cell to cell. Misfolded α-syn fibrils can induce monomeric α-syn to misfold in the cell and are released into the extracellular space where they can further enter neighboring cells to seed soluble α-syn into a misfolded and aggregated form. Preformed α-syn fibril(s) (PFF) can be injected into the striatum or substantia nigra (SN) using animal models for the development of Lewy-like α-syn fibrillar inclusions and aggregates that closely recapitulate features of human PD [25]. The possibility of combining different genetic-based models of α-syn pathology has also been explored; Thakur et al. have injected exogeneous PFF into the SN and ventral tegmental area (VTA) of the rat brain that overexpressed AAV-derived human α-syn to further speed up the process of α-syn fibrillar inclusions [25].

Interestingly, recent studies have provided direct evidence of gut-to-brain α-syn transmission in rodent [26] and mouse [27] models, supporting the classical Braak’s hypothesis that sporadic PD could be caused by pathogen (virus or bacterium) entering the gut via the nasal cavity [28]. Such trans-neuronal propagation of pathologic α-syn was further accompanied by loss of DaNs and behavioral deficits [27], providing novel PD models to explore the role of gut–brain axis in PD pathogenesis. In a similar manner, Van Den Berge et al. found evidence of transmission of α-syn pathology through both sympathetic and parasympathetic pathways from the duodenum to the dorsal motor nucleus of the vagus (DMV) and locus coeruleus in transgenic rats that overexpressed a human form of α-syn [29]. While these new models may be useful in studying specific cellular and molecular pathways in PD and related α-synucleinopathies, it takes a substantially long time (>6 months) for significant neurodegenerative changes to appear in midbrain DaNs following the injection of PFF into the gut. Here we discuss recent sc/snRNA-seq efforts that enabled single-cell characterization of animal and human in vitro models of PD and human postmortem brain tissues obtained from controls and/or PD patients (Table 1).

### 2.1. Mouse SN-Derived DaNs

The earliest attempts at elucidating the extent of DaN development and diversity through the analysis of sc/snRNA-seq used mouse embryos and early postnatal DaNs, resolving temporal and spatial dynamics and molecularly defined cell types during ventral midbrain development [34,37,38]. In 2016, the integrative analysis of scRNA-seq data derived from ventral midbrain (VM) in human and mouse identified specific adult dopaminergic cell types that emerged postnatally and several diverse radial glia-like cell types biased toward a distinct fate (Figure 1a,b) [38]. This study faithfully depicted the degree to which species differ in developmental timing and cell proliferation in the context of molecular diversity.

In line with this study, scRNA-seq of proliferating mesencephalic DaN neural progenitor cells in Lmx1a^EGFP^ mice from four different embryonic days showed a strong axis of differentiation (Figure 1c–e). In-depth analysis of the generated dataset revealed a close association between developing DaNs and subthalamic nucleus neurons, while identifying a specific set of unique transcription factors that can classify the two neuronal subtypes [37]. In subsequent years, the same research group followed later maturation by sampling both postmitotic DaN precursor cells and differentiated neurons from the mouse VM at six different time points during DaN maturation from E13.5 to postnatal day (P) P90 in Pitx3^eGPF/wt^ mice (Figure 1f,g). Validated by histological analysis, the network analyses identified seven neuronal subpopulations divided into two major branches of Pitx3-expressing neurons differing in the expression of Slc6a3, revealing novel cellular populations that are developmentally related but are non-dopaminergic [34]. Altogether, these studies have important implications for developing a cell therapy strategy for PD, as the refinement and optimization of differentiation protocols will require better understanding of dopaminergic specification, neurogenesis, and diversity.

Mouse brain tissues have been subjected to sc/snRNA-seq not only to elucidate the diversity and development of DaNs, but also to identify putative cell types specifically vulnerable in PD. Through the expression-weighted cell-type enrichment (EWCE) [46] analysis, Bryosis et al. found that genes that were upregulated in brains with higher Braak scores were specifically enriched in oligodendrocytes and not microglia, while downregulated genes were expressed only in DaNs using mouse brain-derived scRNA-seq [36]. DaN-specific expression patterns revealed by sc/snRNA-seq were further used for prioritizing candidate genes in previously identified GWAS loci associated with sporadic PD susceptibility through PD GWAS loci gene scoring [33]. Computational approaches used in these studies are discussed in detail in Section 4.2.

### 2.2. Human iPSC/Embrionic Stem Cell (ESC)-Derived DaNs

The advances in induced pluripotent stem cell (iPSC) technology have revolutionized our ability to model PD and have brought much success in generating human in vitro neurons that would be otherwise inaccessible. As compared with postmortem tissues representing the endpoint of disease, reprogramming PD patient-derived cells into iPSCs followed by subsequent differentiation into DaNs represents the earliest stages of the disease process, facilitating discovery of novel biomarkers and therapeutic candidates [31,47]. Successful generation of iPSC-derived DaNs from patients harboring PD mutations or alterations in GBA (RecNcil, L444P, N370S) [48,49], SNCA (triplication) [50,51,52], or LRRK2 (G2019S) [53,54,55,56] has elucidated the role of α-syn in the origin and progression of PD.

Further, human iPSC (hiPSC)-derived neuronal models have enhanced the understanding of the role of PD-causing mutations. For example, it has been proposed that LRRK2-G2019S results in dopaminergic neurodegeneration with its functional role in mRNA translation and calcium homeostasis [57], oxidative phosphorylation [58], mitochondrial DNA damage [55], interferon gamma (IFN-γ) signaling [59], neuritogenesis [60], phagosome maturation [61], and lysosomal tubulation and vesicle sorting [62]. Similarly, the roles of SNCA-A53T in axodendritic neuropathology [63], cellular bioenergetics [64], endoplasmic reticulum (ER)-to-Golgi complex trafficking [65], mitochondrial dysfunction, and neuronal apoptosis [66] have been suggested through rigorous exploration of hiPSC-derived models. As familial PD represents <10% of cases, human iPSC lines from patients with young-onset PD with no known PD mutations have also recently been established [47], although such iPSC models of sporadic PD often do not show α-syn accumulation as compared with wild-type (WT) controls [53,67].

Nevertheless, regardless of the midbrain patterning protocol, DaN cultures can be highly heterogeneous. Comprehensive meta-analysis of published hiPSC-PD studies found that proportions of generated neurons and DaNs vary greatly between studies, even when using identical differentiation methods, with 27% of cell populations being DaNs, on average [6]. Further, the presence of other cell types, such as astrocytes, neural stem cells, and glutaminergic and GABAergic neuronal subtypes in cell cultures was often inevitable [6,68,69], adding layers of complexity and inconsistencies in downstream analyses. To date, these cell cultures exhibiting varying degrees of heterogeneity and cellular variability have been explored by bulk-cell approaches that are unable to characterize individual cells.

To overcome these limitations, Lang et al. performed both bulk and single-cell RNA-seq using hiPSC-derived DaNs from controls and PD GBA-N370S patients and revealed a functionally enriched gene set that defined a pseudotemporal axis of gene expression variation in mutant hiPSC-derived DaN [40]. Using ingenuity pathway analysis (IPA), they found that the downregulation of HDAC4-controlled genes occurs early along the axis of disease and that pharmacological modulation of HDAC4 activity or localization rescued PD-related phenotypes, including ER stress and autophagic and lysosomal perturbations, and increased in α-syn release. In another study, droplet-based scRNA-seq of WT hiPSC-derived DaNs identified six distinct cell types, including two neuron progenetic populations expressing dopaminergic progenitor markers (i.e., VIM, HES1, and NFIA) and four DaN populations expressing mature neuronal markers (i.e., MAP2 and SNAP25) and dopaminergic lineage markers (i.e., PBX1, KCNJ6) [31] (Figure 2a–c). The sensitivity to oxidative stress and ER stress was further assessed in a cell type-specific manner using hiPSCs-derived WT DaNs and isogenic SNCA-A53T mutant DaN subpopulations. Overall, this study performed an in-depth scRNA-seq analysis and provided a rich resource (accession code: ArrayExpress E-MTAB-9154) with which to explore cell type-specific responses to PD-relevant stress-induced perturbations.

Stem cells can also be derived from fetal sources and embryonic origins. Their self-renewal ability (i.e., capable of infinite expansion) makes stem cells ideal candidates for cell replacement therapies in PD. The engraftment of human pluripotent stem cell-derived neural progenitors and/or functional neurons have been proven safe and efficient in animal models of PD [70,71,72,73] and even in a PD patient [74]. Nevertheless, to develop a gold standard, and a possibly personalized cell therapy strategy for PD, it is important address some of the issues that may arise. As the field is still in its infancy, the cell manufacturing process should carefully (1) derive the right neural cell type (e.g., caudal midbrain DaNs) or cell state (e.g., progenitor cells, intermediate, or fully differentiated) for transplantation, (2) determine the initial source of stem cells (e.g., fetal, ESC, or iPSC) for immunosuppression, (3) identify the number of cells and site of transplantation, and (4) eliminate the risk of tumorigenesis [75].

Increasingly, scRNA-seq technologies and analyses have been used to examine the safety, efficacy, and reproducibility of the generated neurons that were grafted. For example, the Takahasi research group performed a single-cell RT-qPCR analysis to assess the expression of genes related to DaN differentiation, proving the reproducibility of their results [72]. Through the combined analyses of scRNA-seq and histology, Tiklová et al. have also recently characterized intracerebral grafts derived from human embryonic stem cells (hESCs) and ventral midbrain (VM) fetal tissue, unraveling previously unknown cellular diversity and composition in a pre-clinical rat PD model [35]. In addition to neurons and astrocytes, a class of newly identified perivascular-like cells was identified as having a novel cellular composition of hESC-derived grafts in this study (Figure 2d–g).

## 3. Human Postmortem Substantia Nigra

Recent efforts to profile individual nuclei from human postmortem brain tissues have demonstrated efficient classification of cell types and/or assessment of spatiotemporal dynamics of cellular compositions at the single-cell resolution [30,32,44,45,76,77,78,79,80,81,82,83]. For example, a survey of human neocortex transcriptome diversity identified novel subpopulations of adult neurons that expressed major histocompatibility complex type I genes, in which such an expression pattern was not observed in fetal neurons [81]. In another study, snRNA-seq of cells obtained from six neocortical regions further identified 16 neuronal subtypes, with distinct interneuron cell populations and excitatory neurons showing unique spatial organization [76]. By combining bulk tissue RNA-seq and scRNA-seq approaches, Liu et al. further profiled lncRNAs, including polyA selected and total RNA, obtained from human neocortex at different stages of development, and identified a specific target enriched in radial glia cells but not in tissues [77]. Consequently, such studies have constructed open-source sc/snRNA-seq databases, including a single-cell atlas of the mid-gestation human neocortex [82] and SN [30] and the Allen Brain Cell Types Database containing the primary motor cortex (M1), multiple cortical areas, middle temporal gyrus (MTG), primary visual cortex (V1C), and anterior cingulate cortex (ACC) [83].

Human single-nucleus transcriptomic atlases for the substantia nigra (SN) have identified cell clusters spanning known resident cell classes, including astrocytes (ASC), oligodendrocytes (ODC), oligodendrocyte progenitor cells (OPC), mural cells (endothelial cells and pericytes), microglia, fibroblasts, and neurons, including DaNs and multiple inhibitory types [30,32]. While most of the nuclei obtained from SN were identified as glial cell populations (95.5%), which mainly comprised ODCs (72%), only 12% of cell populations were glia, with the rest of captured nuclei comprising mainly excitatory (Ex) and inhibitory (In) neurons [30] (Figure 3a). The integration of SN and cortex snRNA-seq data further reveals transcriptional correlation attributed to cell type rather than the region of origin (Figure 3b). Importantly, DaN-specific expression patterns identified from these merged data were associated with established genetic variants contributing to PD traits or genetic risk loci (see Section 4.2. for further details, including computational tools used in the analyses) [30]. LIGER, a computational algorithm that integrates highly heterogeneous sc/snRNA-seq datasets, was successfully applied to analyze a total of 44,274 nuclei derived from the SN of seven healthy individuals [32]. Despite substantial variation observed across different individuals (Figure 3c), LIGER identified 24 cell populations (Figure 3d,e) in which the expression patterns were strongly concordant with cell clusters identified from mouse SN, consistent with another study that observed well-conserved cell types in human and mouse cortex [83].

Importantly, one of the first attempts to profile postmortem brain tissues from idiopathic PD patients at the single-cell resolution has been made recently, providing comprehensive insight into the molecular composition and cellular phenotype of PD [39]. In this study, PD-specific upregulation of microglia and astrocytes was associated with cytokine signaling and the stress response to unfolded proteins, implicating the role of glial cells in the neuroinflammatory process in the disease. In accordance with earlier observations [30,32], the majority of cell populations consisted of glial cells (~80%) while DaNs contributed to 0.18% of the total cell count. These results highlight important technical limitation in current sc/snRNA-seq protocols, which could miss relatively short poly(A) tail of Th mRNA transcripts. Nevertheless, to the best of our knowledge, this was the first study that demonstrated successful enrichment of PD-specific DaNs and glial subpopulations through the comparative analysis of snRNA-seq derived directly from postmortem brain tissues of PD patients.

## 4. Emerging Tools for Data Analysis

Currently, there is a plethora of different sc/snRNA-seq technologies and computational analytical tools, allowing us to create tailored experimental and computational designs for studies. Compared to the first attempt to measure the expression of several genes from a few single cells using in vivo reverse transcription- and in vitro transcription-based approaches in 1992, recently developed in situ barcoding-based methods allow the sequencing of up to several millions of cells (key technological developments are described elsewhere [84]). Multiple rounds of split-pool barcoding of (pre)mRNAs would make such combinatorial indexing strategies more affordable and effective as compared with the existing sc/snRNA-seq technologies that rely heavily on physical compartments (e.g., wells, traps, droplets) to isolate individual cells and generate sequencing libraries on a per-cell basis.

A systemic comparison of existing sc/snRNA-seq technologies, however, revealed notable differences in read structure and alignment efficiency among different methods [85]. Such efficiency was assessed based on several key metrics, such as the presence of poly(T) in reads and antisense reads, the fraction of mitochondrial and nuclear reads, the number of detected UMIs or genes per cell, the detection of multiplets, technical precision, reproducibility between replicates, accuracy in gene expression, and the ability to classify heterogeneous cell populations (refer to the recently published work [85] for detailed analysis). Here we focused on the emerging sc/snRNA-seq experimental and analytical tools that can be applied in studies of PD.

### 4.1. RNA Velocity

In 2018, the concept of RNA velocity was first introduced as an alternative computational approach for inferring the transcriptional dynamics of individual cells based on the ratio of unspliced to spliced mRNAs [86]. It was suggested that 15–25% of reads sequenced using SMART-seq2 [87], STRT/C1 [88], inDrop [89], and 10× Genomics Chromium [90] protocols contain unspliced intronic reads, which could represent nascent mRNAs. This new analytical tool developed for analyzing scRNA-seq data allows prediction of the rate and direction of change in gene expression and tracking of the formation of stimuli-specific gene modules. Newly developed snRNA-seq technologies will particularly advance the analysis of RNA velocity, given that sequencing single nuclei rather than the intact cell could potentially enrich the amount of sequenced unspliced reads from precursor mRNA (pre-mRNA).

Through the analysis of RNA velocity using ectoderm-derived hypothalamic cell pools, specific cell subpopulations, termed “bridge cells” that link progenitors and immature neurons, have been identified [91]. Interestingly, RNA velocity vector embeddings of Th+ cells in this study identified molecularly distinct subtypes of phenotypically uniform neurons that are critical to hypothalamic development. In another study, UMAP plotting combined with RNA velocity allowed delineation of developmental trajectories for all major hypothalamic cell types, revealing the age at which molecularly defined cell types deviate from the expected gene expression (Figure 4a) [92]. The RNA velocity analysis of E11–E13 developing diencephalon further identified four spatially segregated main hypothalamic domains using the reference region-specific markers that were previously defined (Figure 4b). We believe that sc/snRNA-seq datasets derived from DaNs obtained during ventral midbrain development [34,37,38] combined with the RNA velocity analysis will routinely be used in the near future to elucidate the cellular composition and diversity of DaNs.

### 4.2. Combined Analysis of DaN-Specific Gene Expression and GWAS Results

Through sophisticated computational analysis of GWAS results and bulk and/or sc/snRNA-seq data, recent studies have prioritized specific types of central nervous system (CNS) for follow-up experiments for multiple traits (independent risk loci associated with the disease) in PD [30,33,36]. Computational tools developed for GWAS enrichment analysis are presented (and their abbreviations are defined) in Table 2 and include ALIGATOR [93], CytoScape [94], DAPPLE [95], DAVID [96], DEPICT [97], INRICH [98], MAGMA [99], and WGCNA [100], and those for genome-based heritability analysis include GCTA [101], LDAK [102], LDRegress [103], LDSC [104], MEGHA [105], and PCGC [106] (Table 2). Comprehensive software packages and open-source tools developed for GWAS data analysis include BEAGLE [107], EIGENSOFT [108], Genetic Power Calculator [109], LocusZoom [110], METAL [111], Minimac [112], and PLINK [113]. A combination of these tools can be used in sc/snRNA-seq studies to connect human genomic PD findings to specific brain cell types and provide significant insight into the etiology of PD.

Most commonly used computational tools for SNP-based heritability analysis and enrichment analysis of GWAS data in recent studies are Multi-Marker Analysis of Genomic Annotation (MAGMA) [99] and LD Score Regression (LDSC) [104], respectively. Leveraging these methods, Agarwal et al. found a significant association between PD genetic risk and DaN-specific gene expression patterns, which were identified from snRNA-seq using the SN brain tissues, for the first time in humans (Figure 4c) [30]. This finding is consistent with observations made using mouse-derived expression data of predicted PD GWA-validated risk variants [12,36]. Such an exemplary study demonstrated the ability of sc/snRNA-seq technologies to unravel the heterogeneity of complex brain tissue and reveal potential contribution of SN to PD in a cell type-specific manner.

Despite the central focus on nigrostriatal DaNs in the field, however, there is emerging evidence to suggest the involvement of glial cell populations and, more specifically, astrocytes, microglia, and oligodendrocytes in the pathogenesis of PD [30,36,114,115,116]. By integrating previously published GWAS summary statistics with single-cell transcriptomic data from the entire mouse nervous system and by employing MAGMA [99] and LDSC [104], Bryois et al. recently found that PD was independently associated with oligodendrocytes and enteric neurons, in addition to the cholinergic and monoaminergic neurons (e.g., DaNs) [36]. In support of this view, using MAGMA [99] and LDSC [104], Reynolds et al. reported that the enrichment of PD heritability was observed in a lysosomal-related gene set highly expressed in astrocytes, microglia, and oligodendrocyte subpopulations and not in brain-related cell type-specific annotations [116]. PD-associated risk variants were further found to be associated with lymphocytes, mesendoderm, liver-cells, and fat-cells [117], in addition to being associated with the adaptive and the innate immune system [116,118,119,120]. These studies altogether highlight the need to move beyond assessing only the brain and selective neuronal vulnerability.

### 4.3. Machine Learning Approaches

Cell type annotation is a vital step for subsequent analyses in the computational pipeline for sc/snRNA-seq. Commonly, prior knowledge of established cell type-specific markers (scCATCH [121]) or reference databases of bulk or sc/snRNA-seq profiles (CHETAH [122], scHCL [123], scMap [124], SingleR [125]) are used to annotate each cell type. Increasingly, machine learning-based reference-dependent (CellAssign [126], Garnett [127]) and reference-free (scDeepSort [128]) algorithms have emerged as powerful tools that can rapidly and accurately label cells without prior reference knowledge. In sc/snRNA-seq studies of PD, a stratified cross-validation machine learning approach was implemented to validate manually annotated cell clusters [39]. Further, supervised machine learning was used in a scRNA-seq study of mouse and human ventral midbrain development to assess the quality of the in vitro-differentiated cells relative to in vivo-defined cell types [38]. Thus, we believe that advances in deep learning along with sc/snRNA-seq technologies will be particularly valuable to stem cell research and in silico biomarker discovery in studies of PD.

### 4.4. Challenges and Prospects

The central focus in the field has been the ability to identify known and novel cell types in an unbiased and efficient manner. However, different experimental and/or computational tools and parameters choices can readily lead to disparate outcomes for the same sc/snRNA-seq dataset. The performance of existing sc/snRNA-seq technologies further varies between cell types and samples of different tissues and species, affecting reproducibility and downstream functional interpretation. For example, a recent multicenter benchmark study comparing 13 commonly used scRNA-seq protocols, including CEL-seq2 [129], MARS-seq [130], Quartz-seq2 [131], gmcSCRB-seq [132], Smart-seq2 [87], ddSEQ (Bio-Rad), ICELL8 [133], C1 High-Throughput (C1HT-small, C1HT-medium) [134], 10 × Chromium [90], Drop-seq [135], and inDrop [89], highlighted differing library complexity and their cell mapping predictive values by analyzing human peripheral blood and mouse colon tissue [136]. In this section, important drawbacks that are common to the existing sc/snRNA-seq technologies were highlighted, with more technical details and a possible solution for addressing such issues through sophisticated analytic strategies.

#### 4.4.1. RNA Postmortem Degradation

Studies of the human brain have been limited due mainly to difficulty in acquiring postmortem samples and to the low quality associated with RNA degradation. Optimal postmortem interval (i.e., the time that has elapsed between the subject’s death and processing of tissue) may be determined prior to conducting the actual sc/snRNA-seq experiments depending on the biomolecules of interest, including messenger RNA (mRNA), microRNA (miRNA), histone modifications, and proteins. For example, Nagy et al. found that while miRNA was resistant to postmortem intervals, histone modifications and proteins showed a threshold between 72 and 96 h [137]. Postmortem interval-related mRNA degradation was found to be transcript-, tissue-, and gene-specific [137,138,139]. Careful consideration of the assay design, sample preparation, and experimental protocols are thus essential for accurate downstream analyses for sc/snRNA-seq research.

#### 4.4.2. Doublets

Prior to library preparation and sequencing, complete mechanical dissociation of brain tissue into single-cell suspensions should be achieved to avoid generation of “doublets”—two or more cells that share the same molecular tags or barcodes. Failed removal of doublets could potentially lead to nonexistent expression profiles that misguide downstream analyses, including dimensionality reduction, cell type identification, differential expression, and trajectory inference [140]. Generation of pure single-cell suspensions from brain tissues, however, remains experimentally challenging mainly because of mature neurons containing axonal projections. Notably, cells of neuronal lineages were frequently underrepresented among sequenced cell populations, even in studies leveraging recent sc/snRNA-seq protocols [30,35,39,41].

Experimental approaches used to address such issues have involved loading of cells at low concentrations and “hashing” of the cells with barcoded antibodies [141] or multiplexing using lipid-tagged indices [142]. While these attempts could minimize the occurrence of doublets, they are costly and require additional materials and laborious procedures. Alternatively, doublets could be detected computationally and filtered out during a QC preprocessing step through machine learning. Such analytical tools, including DoubletFinder [143], Scrublet [144], and scds [145], develop and train a classifier to identify doublets based on the profiled sc/snRNA-seq data derived from mixed cell populations comprising mixed singlets and doublets. The number of unique genes, or UMIs, should also be assessed to detect and remove potential doublets or multiplets exhibiting an aberrantly high gene count. Nevertheless, these approaches should carefully be implemented in the integrated workflow when the identified cell clusters are not fully defined.

#### 4.4.3. Study or Batch Effects

Sc/snRNA-seq data are often derived and compiled from various experiments, including library preparation and sequencing, with differences in sc/snRNA-seq protocol (e.g., cell or nuclei extraction, fixation, permeabilization, reverse transcription, barcode ligation, double stranded cDNA synthesis, and/or tagmentation), reagents, experimental duration, handling personnel, and sequencing platforms. Despite systemic differences and resulting batch or study effects, computational tools, mostly available in the R packages, allow batch- or study-effect correction and generation of an integrated gene expression matrix for downstream analysis [10].

For example, the *align_cds* function in a newly updated R package *monocle3* (v0.2.2)—an analysis toolkit for sc/snRNA-seq—allows the removal of batch effects using mutual nearest neighbor alignment by calling the *mnnCorrect* function implemented in another R package, *batchelor* (v1.6.2) [146]. Similarly, the *FindIntegrationAnchors* and *IntegrateData* functions implemented in the R package *Seurat* (v3) integrate multiple distinct scRNA-seq datasets produced across different scRNA-seq technologies [147]. These functions aim to identify “anchors” between pairs of scRNA-seq datasets, representing pairwise correspondences between individual cells. Here the standard preprocessing of the datasets, including log-normalization and the identification of variable features, should be performed prior to finding anchors. The *Seurat* v3 further supports the projection of the integrated reference data (meta-data) onto the query expression data, facilitating efficient cell type classification that can further be validated using known canonical cell type markers. Other commonly used computational tools for correcting study and/or batch effects across different sc/snRNA-seq datasets include the tools developed for microarray data batch correction, such as ComBat [148] and limma [149].

While these batch-effect correction methods have been widely recognized as efficient computational approaches for integrating multiple independent datasets, each method had its advantages and limitations in terms of computational runtime, memory usage, batch integration capability, and the ability to handle large datasets, detect differentially expressed genes, and identify correct cell type (refer to recent work [150] for the benchmark study on 14 batch-effect correction methods; Harmony [151], LIGER [32], and Seurat 3 [147] were found to be the top batch mixing methods in this study). It is thus recommended that the performance of various batch correcting algorithms should be assessed with the datasets of interest prior to establishing a computational pipeline for sc/snRNA-seq analysis.

## 5. Concluding Remarks

Rapid progress in the development of sc/snRNA-seq technologies along with analytical tools has greatly advanced the understanding of the molecular identity of neuronal and glial cell types in studies of PD. In this review, we aimed to review the sc/snRNA-seq technologies and their application to PD that were reported over the past five years. Emerging computational technologies including deep learning and spatially resolved technologies for SN profiling as well as the integration of omics data from disparate sources and technologies have revealed PD-specific, previously unknown subpopulations of DaNs and glia, and have generated valuable single-cell atlases that could serve as reference data for future research. Considering that mechanism-based treatments still remain elusive for PD, we believe that continuous improvement in both experimental and analytical tools developed for sc/snRNA-seq will dramatically advance the ability to develop novel gene-based biomarkers for diagnosis, prognosis, and targeted therapy in PD.

## Figures and Tables

**Figure 1 biomedicines-09-00368-f001:**
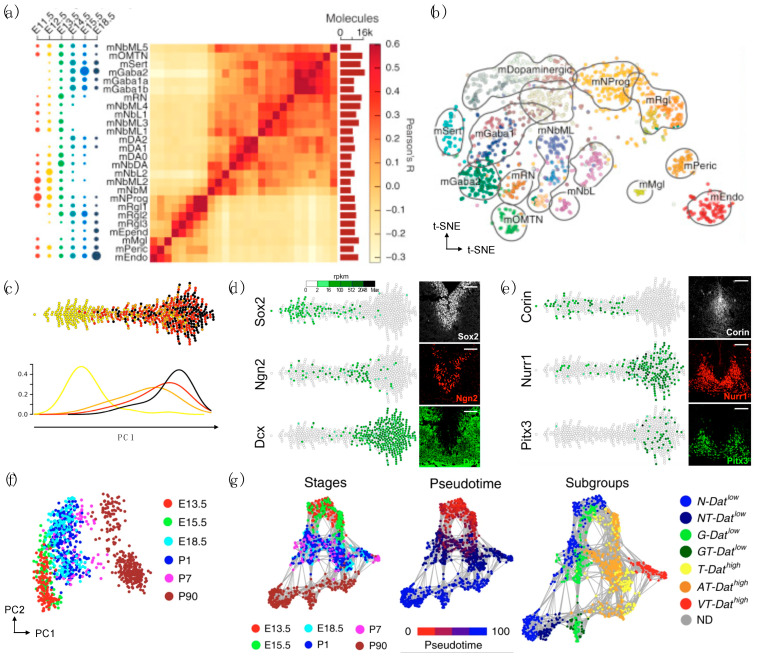
scRNA-seq of mouse neural progenitors from embryos and early postnatal DaNs. (**a**,**b**) Use of ventral midbrain at six embryonic (E) days from E11.5 to E18.5 in mouse embryos. Reprinted from [38], Copyright 2016 Elsevier. (**a**) Dot plot depicting time distribution of cell types (left), heatmap showing pairwise correlation (middle), and bars showing average number of mRNA molecules per cell (right). (**b**) t-distributed stochastic neighbor embedding (t-SNE) of cells colored by cell type. (**c**–**e**) Use of mesencephalic DaNs at four embryonic (E) days from E10.5 to E13.5 in Lmx1a^EGFP^ mice. Reprinted from [37], Copyright 2017, with permission from Elsevier. (**c**) Cells plotted along the first principal component (PC1), colored by embryonic days (top) and the frequency distribution (bottom); yellow: E10.5, orange: E11.5, red: 12.5, and black: E13.5. Relative expression of (**d**) pan-neuronal markers and (**e**) DaN markers along PC1 (left) and co-immunostainings of the stated markers (right). (**f**,**g**) Use of ventral midbrain at three embryonic (E) days from E13.5 to E18.5 and three postnatal (P) days from P1 to P90 in Pitx3^eGPF/wt^ mice. Reprinted from [34]. Copyright 2019, Katarína Tiklová et al. (**f**) Principal component (PC) plot showing 1106 cells colored by developmental stage. (**g**) Network plot depicting distribution of Pitx3-expressing midbrain neurons colored by developmental stage, pseudotime, and molecularly defined cell type. Dat: Slc6a3, T: Th, N: Nxph4, G: Gad2, A: Aldh1a1, and V: Vip.

**Figure 2 biomedicines-09-00368-f002:**
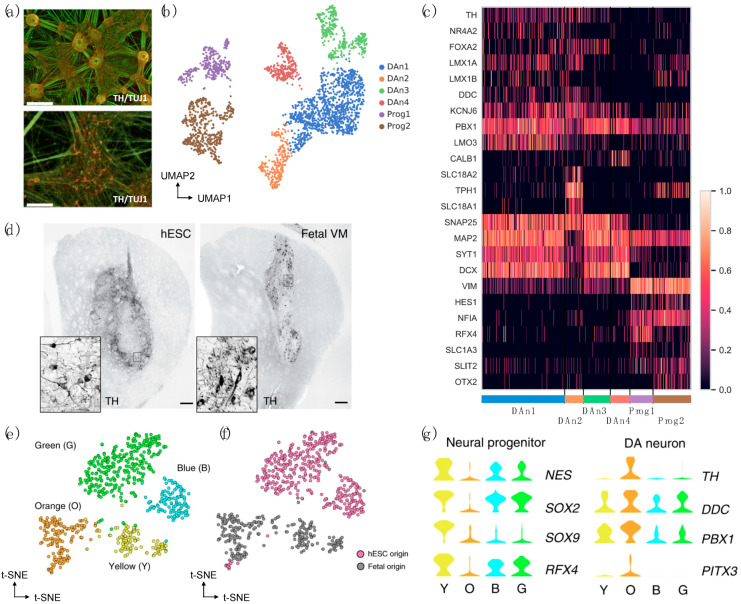
scRNA-seq of fetal and human induced pluripotent stem cell (hiPSC) or embryonic stem cell (ESC)-derived neurons in studies of PD. (**a**–**c**) Use of WT hiPSC-derived DaNs. Reprinted from [31], Copyright 2020 Elsevier. (**a**) Immunofluoresence staining, (**b**) Uniform manifold approxiamation and projection (UMAP) and (**c**) expression heatmap of WT hiPSC-derived DaNs. TH: tyrosine hydroxylase, TUJ1: beta-3-tubulin. (**d**–**g**) Use of human embryonic stem cell (hESC) and fetal ventral midbrain (VM)-derived progenitors. Reprinted from [35], Copyright 2020, Katarína Tiklová et al. (**d**) Immunohistochemistry of TH in the graft core (six months post-transplantation) (**e**,**f**) t-SNE and (**g**) expression plot using the stated genes (before grafting). “Y”, “O”, “B”, and “G” indicate cell clusters shown in (**e**).

**Figure 3 biomedicines-09-00368-f003:**
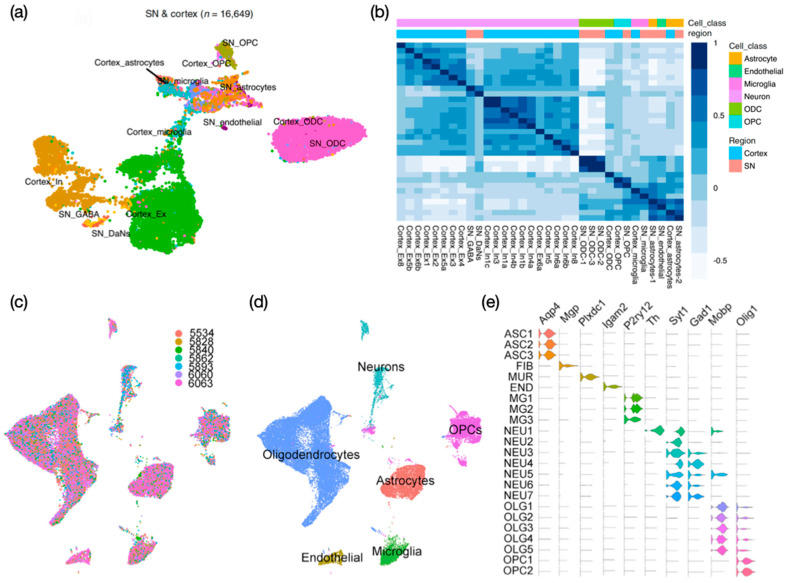
The snRNA-seq of human postmortem brain tissues in studies of PD. (**a**,**b**) Use of SN and cortex tissues derived from five healthy individuals. Reprinted from [30], Copyright 2020, Devika Agarwal et al. (**a**) UMAP (colored by cell type) and (**b**) correlation heatmap depicting hierarchical clustering with Pearson correlation as distance metric. (**c**–**e**) Use of SN tissues derived from seven healthy individuals. Reprinted from [32], Copyright 2019, with permission from Elsevier. UMAP plots colored by (**c**) donor and (**d**) cell type. (**e**) Violin plots depicting expression of PD-related genes across the identified cell types.

**Figure 4 biomedicines-09-00368-f004:**
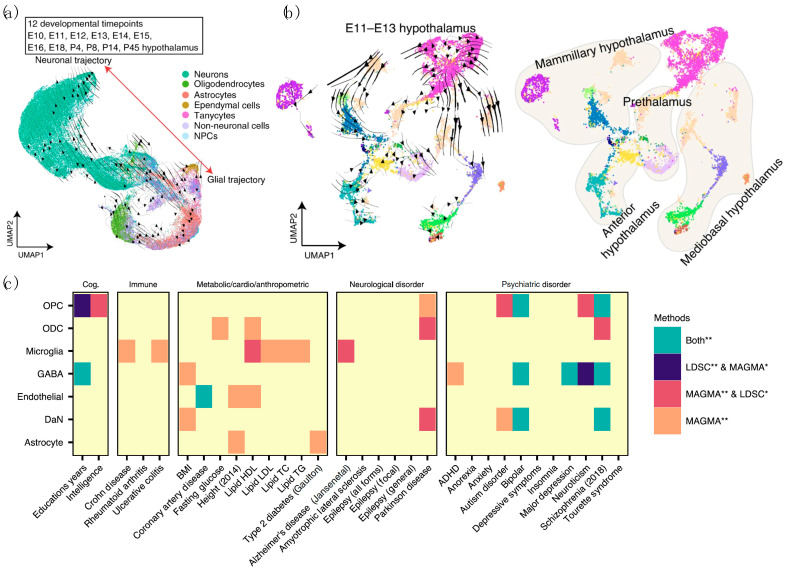
Emerging computational and analytical tools that can be used in sc/snRNA-seq studies of PD. UMAP and RNA velocity trajectories of cells from (**a**) E10–P45 and (**b**) E11–E13 developing diencephalon in mice. Reprinted from [92], Copyright 2020, Dong Won Kim et al. (**c**) Use of LD Score Regression (LDSC) and Multi-Marker Analysis of Genomic Annotation (MAGMA) to assess the associations between previously reported genetic risk variants of different complex “traits”, or brain-related disorders, and SN cell types. Reprinted from [30], Copyright 2020, Devika Agarwal et al. Color of heatmap indicates degrees of statistical significance. *: *p* value (<0.05), **: Bonferroni corrected q value, Cog.: cognitive phenotypes, Immune: autoimmune diseases, Metabolic/Cardio/Anthropometric: metabolic, cardiovascular and anthropometric traits.

**Table 1 biomedicines-09-00368-t001:** Summary of single-cell and single-nucleus RNA sequencing (sc/snRNA-seq) datasets analyzed in studies of Parkinson’s disease (PD) and in those of dopaminergic neuron (DaN) neurogenesis.

Sample Origin	Condition	Brain Region	Number of Single Cell	Cell or Nucleus	Number of Cell Cluster	sc/snRNA-seq Technology	Reference
Human postmortem	Wild-type (WT)	SN, cortex	17,000	Nucleus	SN: 10, cortex: 6	10×	[30]
Human iPSC	WT, oxidative stress-induced, SNCA-A53T mutant	-	15,325	Cell	WT: 6	10×	[31]
Human postmortem	WT	SN	44,274	Nucleus	24	10×	[32]
Mouse tissue	WT	Midbrain, forebrain, olfactory bulb ^1^	396	Cell	13	Smart-seq2	[33]
Mouse tissue	WT	Ventral midbrain ^2^	1106	Cell	8	Smart-seq2	[34]
Rat tissue	PD model	Striatum (Str), midbrain (mid)	Str: 746, mid: 7875	Cell	4	Smart-seq2, 10×	[35]
Mouse tissue ^4^	WT	Entire nervous system	509,876	Cell	B: 39, R: 265 ^3^	10×	[36]
Mouse tissue ^5^	WT	9 brain regions ^6^	690,000	Cell, nucleus	565	Drop-seq	
Mouse tissue ^7^	WT	5 brain regions	~10,000	Cell, nucleus	B: 24, R: 149 ^8^	DroNc-seq	
Human postmortem ^9^	WT	Hippocampus, prefrontal cortex	19,550	Nucleus	16	DroNc-seq	
Human postmortem ^10^	WT	Visual cortex, frontal cortex, cerebellum	36,166	Nucleus	35	snDrop-seq	
Mouse embryo	WT	Ventral mesencephalic and diencephalic (VMD) region	550	Cell	4 ^11^	Smart-seq2	[37]
Mouse embryo	WT	Ventral midbrain	1907	Cell	26	C1-STRT	[38]
Human postmortem	WT, idiopathic PD patients	Midbrain	41,435	Nucleus	12	10×	[39]
Human iPSC	WT, PD GBA-N370S patients	-	146	Cell	6 ^12^	Smart-seq2	[40]

^1^ Of Th:EGFP BAC transgenic (Tg(Th-EGFP)DJ76Gsat/Mmnc) mice from embryonic day (E) E15.5 and postnatal day (P) P7. ^2^ Of Pitx3^eGPF/wt^ mice from E13.5, 15.5, 18.5, and P1, 7, and 90. ^3^ B refers to broad categories (level 4), and R refers to refined cell types (level 5), respectively. ^4^ This scRNA-seq dataset was generated from [41]. ^5^ This sc/snRNA-seq dataset was generated from [42]. ^6^ Frontal cortex, striatum, globus pallidus externus/nucleus basalis, thalamus, hippocampus, posterior cortex, entopeduncular nucleus/subthalamic nucleus, substantia nigra/ventral tegmental area, and cerebellum. ^7^ This sc/snNRA-seq dataset was generated from [43]. ^8^ B and R refers to broad (level 1) and refined (level 2) cell types, respectively. ^9^ This snRNA-seq dataset was generated from [44]. ^10^ This snRNA-seq dataset was generated from [45]. ^11^ By embryonic age: E10.5, 11.5, 12.5, and 13.5. ^12^ By sample origin: Control1, 2, 3, and PD GBA-N370S patients GBA1, 2, and 3. SN, substantia nigra.

**Table 2 biomedicines-09-00368-t002:** Commonly employed computational and analytical tools in genome-wide association studies (GWAS) and sc/snRNA-seq studies.

Tool	Full Name	Analysis	Feature	Ref.
ALIGATOR	Association List Go Annotator	Pathway analysis tool for GWAS data	Adjust for common genomic confounding factors using well-controlled type I error	[93]
CytoScape	CytoScape	Visualization tool for network and pathway findings	Visualize results for network structure analyses, network clustering, hotspot detection, and functional enrichment	[31,94]
DAPPLE	Disease Association Protein-Protein Link Evaluator	Network-assisted analysis tool for prioritizing GWAS results	Find physical connectivity among proteins encoded by genes in loci associated with disease	[95]
DAVID	Database for Annotation, Visualization, and Integrated Discovery	Pathway analysis tool high-throughput gene-based data	Facilitate functional annotation and analysis of any given list of genes	[96]
DEPICT	Data-Driven Expression-Prioritized Integration for Complex Traits	Integrative GWAS analysis tool	Prioritize most likely causal genes using both established annotations and gene expression data	[97]
GCTA	Genome-Wide Complex Trait Analysis	SNP-based heritability analysis	Estimate the proportion of phenotypic variance explained by whole-genome genotype data	[101]
INRICH	Interval Enrichment Analysis	Pathway analysis tool for GWAS data	Detect enriched association signals of LD-independent genomic regions within biologically relevant gene sets	[98]
LDAK	Linkage Disequilibrium Adjusted Kinships	SNP-based heritability analysis	Create kinship matrices take into account LD between genotype markers	[102]
LDregress	LDregress ^1^	SNP-based heritability analysis	Adjust for LD between genotype markers using regression	[103]
LDSC	LD Score Regression	SNP-based heritability analysis	Use association summary statistics instead of genotype data	[104]
MAGMA	Multi-Marker Analysis of Genomic Annotation	Gene- and generalized gene-set analysis for GWAS data	Analyze both raw genotype data and summary SNP *p*-values from a previous GWAS or meta-analysis	[99]
MEGHA	Massively Expedited Genome-Wide Heritability Analysis	SNP-based heritability analysis	Estimate measures of heritability with several orders of magnitude less time than existing methods	[105]
WGCNA	Weighted Gene Co-Expression Network Analysis	Gene-expression data analysis	Find clusters of highly correlated genes and enriched biology or functions using module eigengenes or intramodular hub genes	[100]

^1^ It is implemented in the EIGENSOFT software.

## Data Availability

Not applicable.

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
