# Peer review of "Single-Cell RNA Sequencing in Parkinson’s Disease"

_biomedicines, 2021, doi:10.3390/biomedicines9040368_

Round 1

Reviewer 1 Report

The authors reviewed recent progress on understanding the molecular pathogenesis of PD with single-cell and single-nucleus RNA sequencing technologies, including studies on in vitro PD models and human post-mortem samples. The authors further discussed the emerging technologies and the limitation and technical challenges.

The review is well organized and informative. It would be better to include more detailed information in the last part of the introduction on single-cell and single-nucleus RNA sequencing technologies, such as a brief introduction of this technology, the advantages and major contributions of applying this technology to PD studies, the even greater potentials to be achieved with emerging technologies.

Author Response

We thank the reviewer for their time and suggestions for the improvement.

Following the reviewer’s comments, we have newly added line 53-69 in the last part of the introduction on sc/snRNA-seq technologies. We believe that this suggestion has been very helpful in improving the quality of the manuscript.

Reviewer 2 Report

The review manuscript ScRNAseq in PD is well written and comprehensively details recent literature work on Single-cell Nucleus and RNAseq on PD. It details various novel findings on DaNs from Sc/SnRNaseq on the mouse model. The authors discussed briefly the technical bottlenecks of single-cell analyses and this review will help researchers who want to work on single-cell aspects in the PD. I recommend this article for publication.

Author Response

We thank the reviewer for their time and positive feedbacks.

Reviewer 3 Report

Minor revision

Single cell or single nucleus RNA sequencing is growing fast and more significant in cracking the genetic causes and biomarker searching in neurogenerative diseases including Parkinson’s disease (PD). The authors summarized the in vitro PD models, development of data analysis in sc/snRNA sequencing and their applications in identification of new biomolecules in PD with a collection of good, sufficient citations. The discussed analytical tools and technology are envisioned to have the potential to be widely used in sc/snRNA-seq field. The comprehensive review of sc/snRNA sequencing and its relevance in PD is well organized and great for the readership of Biomedicines. I would like to recommend publication of this literature review in Biomedicines after the minor suggestions detailed below have been addressed.

  1. In Abstract, past tense should be used for the summarization of the review paper instead of the present tense. The same issue also frequently appears in the main context.

  1. There are many typos and grammatical errors. For instance, “… gens…” should be corrected to “genes”.

It had better be “…is a prion-like disorder…” instead of “…is prion-like disorder…”.

Also, “…entered in the gut…” to “… entering the gut…”.

“…pre-mNRA…” to “…pre-mRNA…”

The English writing needs double checked further before publishing.

  1. In the discussion of the current challenges of sc/sn RNA sequencing, why did the author decide to choose “Study and/or Batch effects” instead of “Study or Batch effects”? Meanwhile, the study or batch effects were discussed to relate to the dissociation bias, methods used in the sample preparation, sequencing designs but the meticulous selection of analytical tools could help. Rare and heterologous tissue samples are also significant variables in scRNA-seq. To the best knowledge of the authors, could the analytical tools discussed in this part also be a practical solution to the sample issue mentioned above?

  1. The list of computational approaches for data analysis is of great help to readers and significance of this paper. However, as for authors, which computational approach is good for PD analysis considering biomarkers searching? Besides, it is suggested that the discussion of specific biomarkers discovered by sc/sn RNA seq and then used in diagnosis of PD or other neurodegenerative diseases like Alzheimer’s disease would further validate the future application of sc/sn RNA-seq in PD, if there is any.

  1. The numbering style of each graph in the figures does not match well with each other which need more revisions. Meanwhile, in Figure 2g, the authors need to annotate “Y”, “O”, “B”, “G” respectively in the figure caption.

Author Response

Single cell or single nucleus RNA sequencing is growing fast and more significant in cracking the genetic causes and biomarker searching in neurogenerative diseases including Parkinson’s disease (PD). The authors summarized the in vitro PD models, development of data analysis in sc/snRNA sequencing and their applications in identification of new biomolecules in PD with a collection of good, sufficient citations. The discussed analytical tools and technology are envisioned to have the potential to be widely used in sc/snRNA-seq field. The comprehensive review of sc/snRNA sequencing and its relevance in PD is well organized and great for the readership of Biomedicines. I would like to recommend publication of this literature review in Biomedicines after the minor suggestions detailed below have been addressed.

  1. In Abstract, past tense should be used for the summarization of the review paper instead of the present tense. The same issue also frequently appears in the main context.

We thank the reviewer for their time and careful review. We have made correction in the abstract and in the main text, as stated below:

Line 10: “..is” to “has been”

Line 15: “ we reviewed” to “we reviewed”

Line 16: “we also examine” to “..examined”

Line 18: “we highlight” to “we highlighted”

Line 85: “..involve” to “have involved”

Line 88: “exhibit” to “exhibited”

Line 89: “lack” to “lacked”

Line 93: “lead” to “led”

Line 122: “we discuss” to “we discussed”

Line 145: “have used” to “used”

Line 150: “depict” to “depicted”

Line 193: “are” to “have been”

Line 199: “can be” to “were”

Line 227: “has found” to “found”

Line 230: “is” to “was”

Line 257: “perform” to “performed”

Line 257: added “provided”

Line 351: “is” to “was”

Line 352: “demonstrates” to “demonstrated”

Line 372: “allow” to “allowed”

Line 379: “reveal” to “revealed”

Line 381: “has been” to “was”

Line 386: “focus” to “focused”

Line 391: “has been” to “was”

Line 454: “demonstrates” to “demonstrated”

Line 467: “have been” to “were”

Line 498: “has shown” to “highlighted”

Line 501: “are” to “were”

Line 509: “have found” to “found”

Line 523: “are commonly” to “were frequently”

Line 526: “include” to “have involved”

  1. There are many typos and grammatical errors. For instance, “… gens…” should be corrected to “genes”.

It had better be “…is a prion-like disorder…” instead of “…is prion-like disorder…”.

Also, “…entered in the gut…” to “… entering the gut…”.

“…pre-mNRA…” to “…pre-mRNA…”

The English writing needs double checked further before publishing.

 We have corrected the above typos and grammatical errors. The English writing has also been checked again thoroughly.

  1. In the discussion of the current challenges of sc/sn RNA sequencing, why did the author decide to choose “Study and/or Batch effects” instead of “Study or Batch effects”? Meanwhile, the study or batch effects were discussed to relate to the dissociation bias, methods used in the sample preparation, sequencing designs but the meticulous selection of analytical tools could help. Rare and heterologous tissue samples are also significant variables in scRNA-seq. To the best knowledge of the authors, could the analytical tools discussed in this part also be a practical solution to the sample issue mentioned above?

We thank the reviewer for their careful review and suggestion. We have changed the term to “Study or Batch Effects” in line 538.

We thank the reviewer for the query. We believe that such analytical and computational tools developed for batch effect correction could, in part, represent a practical solution to the issue mentioned above. These batch effect correction methods have widely been applied in numerous studies, and have successfully integrated multiple sc/snRNA-seq datasets derived from different experimental protocols and assay designs. Nevertheless, there is a need for a unified approach for normalization and imputation as capture efficiencies tend to vary between experimental batches (line 537-546).  Following this query, we have re-written line 543-546 and added reference [10].

  1. The list of computational approaches for data analysis is of great help to readers and significance of this paper. However, as for authors, which computational approach is good for PD analysis considering biomarkers searching? Besides, it is suggested that the discussion of specific biomarkers discovered by sc/sn RNA seq and then used in diagnosis of PD or other neurodegenerative diseases like Alzheimer’s disease would further validate the future application of sc/sn RNA-seq in PD, if there is any.

We thank the reviewer for their comment and query. In the studies of PD, MAGMA and LDSC have frequently been used for combined analysis of GWAS results and sc/snRNA-seq data, possibly due to their superior analytical performance demonstrated in other studies. Their use is highlighted in lines 447 to 470.

The reviewer has asked an important question regarding the use of sc/snRNA-seq-derived biomarkers in diagnosis of PD. Unfortunately, to the best of our knowledge, sc/snRNA-seq technologies are just beginning to be applied in studies of PD, particularly human postmortem brain tissues. Among the studies reviewed in our work (Table 1), only reference #39 used postmortem brain tissues from idiopathic PD patients, while the rest of the studies have examined only the WT samples or animal models of PD, lacking their actual application in diagnosis of PD patients. Accordingly, we have added line 60-63 in the last part of the introduction.

  1. The numbering style of each graph in the figures does not match well with each other which need more revisions. Meanwhile, in Figure 2g, the authors need to annotate “Y”, “O”, “B”, “G” respectively in the figure caption.

We thank the reviewer for their suggestion to revise the figure caption. To avoid confusion, we have re-written the caption in the following order for all figures:

Figure #. Main title of figures. Title of figures from the same study. Copyright information. Caption for each figure.

We thank the reviewer for the query on “Y”, “O”, “B”, “G” annotation – they are stated in Figure 2e (shown in the figure). To avoid confusion, we have added line 265-266.